# MATE: Benchmarking Multi-Agent Reinforcement Learning in Distributed Target Coverage Control

**Xuehai Pan**[1,2,5]
xuehaipan@pku.edu.cn

**Mickel Liu**[1,2,5]
mickelliu@stu.pku.edu.cn

**Fangwei Zhong**[3,5,†]
zfw@pku.edu.cn

**Yaodong Yang**[3,4,5,†]
yaodong.yang@pku.edu.cn

**Song-Chun Zhu**[3,4,5,6]
sczhu@bigai.ai

**Yizhou Wang**[1,2,4]
yizhou.wang@pku.edu.cn

[1] School of Computer Science, Peking University
[2] Center on Frontiers of Computing Studies, Peking University
[3] School of Intelligence Science and Technology, Peking University
[4] Institute for Artificial Intelligence, Peking University
[5] Beijing Institute for General Artificial Intelligence (BIGAI)
[6] Department of Automation, Tsinghua University

## Abstract

We introduce the Multi-Agent Tracking Environment (MATE), a novel multi-agent environment simulates the target coverage control problems in the real world. MATE hosts an asymmetric cooperative-competitive game consisting of two groups of learning agents—"cameras" and "targets"—with opposing interests. Specifically, "cameras", a group of directional sensors, are mandated to actively control the directional perception area to maximize the coverage rate of targets. On the other side, "targets" are mobile agents that aim to transport cargo between multiple randomly assigned warehouses while minimizing the exposure to the camera sensor networks. To showcase the practicality of MATE, we benchmark the multi-agent reinforcement learning (MARL) algorithms from different aspects, including cooperation, communication, scalability, robustness, and asymmetric self-play. We start by reporting results for cooperative tasks using MARL algorithms (MAPPO, IPPO, QMIX, MADDPG) and the results after augmenting with multi-agent communication protocols (TarMAC, I2C). We then evaluate the effectiveness of the popular self-play techniques (PSRO, fictitious self-play) in an asymmetric zero-sum competitive game. This process of co-evolution between cameras and targets helps to realize a less exploitable camera network. We also observe the emergence of different roles of the target agents while incorporating I2C into target-target communication. MATE is written purely in Python and integrated with OpenAI Gym API to enhance user-friendliness. Our project is released at https://github.com/UnrealTracking/mate.

## 1 Introduction

The target coverage problem studies the active control of the perception area of a group of agents to track the targets of interest, e.g., wireless sensor networks [1], surveillance camera networks [2, 3], and unmanned aerial vehicle (UAV) networks [4]. It has much real-life significance and received wide applications relating to social well-being, security, and entertainment. For example, smart camera networks can be used for anti-poaching [5, 6], anti-smuggling [7], border security [8] and—for more recreational uses—person-following [9, 10, 11] in filming and ball-tracking [12] in sports events, etc. However, it remains an open challenge to cooperatively control the cameras

36th Conference on Neural Information Processing Systems (NeurIPS 2022).

---

†Equal Advising.

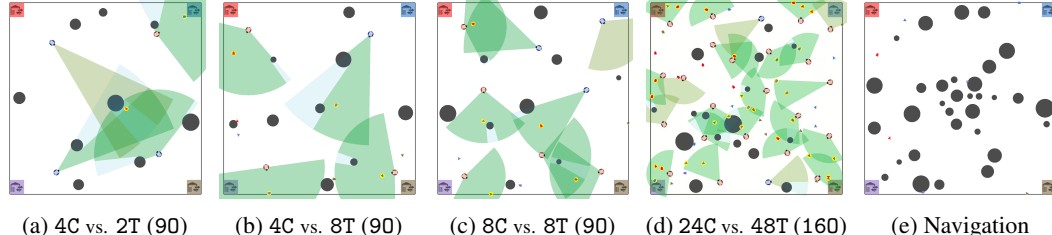

(a) 4C vs. 2T (90)  (b) 4C vs. 8T (90)  (c) 8C vs. 8T (90)  (d) 24C vs. 48T (160)  (e) Navigation

Figure 1: Five snapshots of The Multi-Agent Tracking Environment at different scales. Note that here we abbreviate "camera" as "C", "target" as "T" and "obstacle" as "O".

in distributed networks. A few notable issues hinder the progress: the quantities of the cameras and targets that vary in real-time, the increasingly diverse and unpredictable trajectories of targets, the partial observability of the cameras, and the limited bandwidth of the communication networks. These factors contribute to the difficulties in current research regarding multi-camera cooperation.

Recent successes in multi-agent reinforcement learning (MARL) [13] have demonstrated the superior efficiency of multi-agent learning methods in tackling cooperative-competitive games at super-human levels, as shown in gaming AI [14, 15, 16, 17], robotic manipulation [18, 19], autonomous driving [20, 21], population biology [22], and etc. Assuredly, the research on target coverage problems would benefit significantly from multi-agent reinforcement learning. Unfortunately, we notice that the popular MARL algorithms, e.g., MADDPG [23], QMIX [24], MAPPO [25], HAPPO [26, 27] perform poorly in the target coverage problem [28], though they achieved great success in other existing benchmarks [23, 29, 30]. These benchmarks are either based on video games or some simplified scenarios, neglecting which features the real-world multi-agent applications desperately demand, e.g., heterogeneous agents, asymmetric games, the variable population of agents, simulating partial observation, and peer-to-peer communication. We have yet to see an open-source and standardized environment that benchmarks the MARL algorithms under such practical settings in the context of the target coverage problem.

Motivated by these findings, we build the *Multi-Agent Tracking Environment (MATE)* that advocates the proposal of a more practical multi-agent system. Our system has accounted for both aspects of fully-cooperative and fully-competitive games, the scalability and the robustness of agents, and the communication efficiency among agents. MATE is an open-source simulation that hosts an asymmetric two-team stochastic game between the "cameras" and the "targets". Inside MATE, the camera agents need to maximize the coverage rate on the "targets" while maintaining strong coordination within the team to minimize overlapping detection. On the other hand, the targets are tasked to maximize transport flow between randomly assigned warehouses while minimizing the time of being detected by the cameras. The game-theoretic theme in MATE stimulates the emergence of innovative strategies between two teams of asymmetric agents, thus facilitating the process of autocurriculum. As a design consideration to encourage multi-layer strategies and counterplay, MATE adds randomly spawned obstacles and transport tasks to offset the natural advantages of both types of agents. For example, cameras' mobility is restricted; therefore, targets may hide behind obstacles to temporarily avoid exposure. Still, the freight and bounties would drive the targets to stop hiding as soon as possible and carry on with their assigned tasks.

Worth noting that MATE is not anchored to entirely service the target coverage problem for training a better camera network, but it also stands out as a multi-purpose benchmark to aid the advancement of MARL algorithms. As discussed in the related work section, MATE has many essential features demanded in algorithm-related research in MARL. Besides mixed-motive games, MATE supports fully cooperative and fully competitive game types, which found popularity in the current landscape of theoretical analysis on MARL algorithms [31, 32, 33, 34, 35]. MATE also provides Peer-to-Peer communication channels, a topic that recently gained much interest in developing means beyond information broadcast in multi-agent communication [36, 37]. The competitive game hosted in MATE involves two-team of heterogeneous agents. The target agents differ in carrying capacity and moving speed, marking further heterogeneity in learning the controls of target agents or learning counter strategies against various types of targets as the camera agents.

We showcased a series of experiments to confirm the feasibility of training for both teams inside MATE. We reported the performance of four MARL algorithms (MAPPO [25], IPPO [38], MADDPG [23], QMIX [24]) on different environment configurations. We also showed the performance of our agents in the cooperative game with multi-agent communication add-ons. We observed the emergence of different roles (e.g., "distractors" and "running backs") while training

Table 1: Comparison between relevant MAL environments (viewing with colors is recommended).

| Environment | Game Type | Observations | Actions | Communication | Agent Type | Scalable |
|---|---|---|---|---|---|---|
| MPE (2017) [23] | Mixed-Motive | Continuous | Continuous & Discrete | Broadcast | Heterogeneous | Yes |
| MAgent (2018) [55] | Mixed-Motive | Discrete | Discrete | No | Homogeneous | Yes |
| Pommerman (2018) [56] | Fully-Competitive | Continuous | Discrete | Broadcast | Homogeneous | No |
| MARLÖ (2018) [57] | Mixed-Motive | Continuous + Pixels | Discrete | No | Homogeneous | Yes |
| Hanabi (2019) [58] | Fully-Cooperative | Discrete | Discrete | Broadcast | Homogeneous | No |
| SMAC (2019) [29] | Fully-Cooperative | Continuous | Discrete | No | Heterogeneous | No |
| Neural MMO (2019) [59] | Mixed-Motive | Discrete | Multi-Discrete | No | Homogeneous | Yes |
| GFootball (2019) [60] | Fully-Cooperative | Continuous | Discrete | No | Homogeneous | No |
| MAMuJoCo (2020) [61] | Fully-Cooperative | Continuous | Continuous | No | Heterogeneous | No |
| LBF (2020) [62] | Mixed-Motive | Discrete | Discrete | No | Homogeneous | Yes |
| RWARE (2020) [62] | Mixed-Motive | Discrete | Discrete | Broadcast | Homogeneous | Yes |
| DM Lab2D (2020) [63] | Mixed-Motive | Discrete | Discrete | No | Homogeneous | No |
| Flatland (2020) [64] | Fully-Cooperative | Continuous | Discrete | No | Homogeneous | Yes |
| SMART (2020) [20] | Mixed-Motive | Continuous & Discrete | Continuous & Discrete | No | Heterogeneous | Yes |
| **MATE (Ours)** | Fully-Coop. & Fully-Comp. | Continuous | Continuous & Discrete | Peer-to-Peer | Heterogeneous | Yes |

target agents with communications. Lastly, we employed the adversarial training algorithms (Policy Space Response Oracle (PSRO) [39], fictitious self-play (FSP) [40], and self-play (SP)) in our environment. The results suggest that training with such methods will decrease the exploitability of the trained policies and thus improve their robustness. Regarding the development aspects of MATE, it is extensively integrated with the OpenAI Gym API [41], which enables excellent compatibility with popular RL libraries such as RLlib [42], Tianshou [43], Stable-Baselines-3 [44], and other Gym compatible frameworks. The lightweight of the MATE attributes to efficient computations on CPUs. This property facilitates the opportunities for mass-scale parallelisms using high-throughput architectures like Ape-X [45] and IMPALA [46], etc. Also catering to various research needs, the setting of the environment can also be easily configured, including action space, observability, reward structure, population, and scene layouts. The development team will provide detailed documentation on various features and commit long-term support to this project.

## 2 Related Work

**Target Coverage Problem.** The target coverage problem is to find an optimal control strategy for sensors such that the time to monitor every interested target can be as long as possible [47]. It is a long-standing problem in directional sensor networks [48, 49], robotics [50, 51, 52, 53], and computer vision [3, 54]. Most previous algorithms are heuristically designed for a specific setting or application, lacking a general solution for this problem. Recently, Xu *et al.* [28] built a 2D environment, formulated the problem as a multi-agent cooperative game, and introduced a hierarchical multi-agent reinforcement learning approach to solve this game. However, compared with the real-world scenarios, the environment is over-simplified due to random-walking targets and a lack of obstacles. In MATE, we aim to build a more realistic simulator for benchmarking the off-the-shelf learning algorithms, e.g., account for occlusion caused by obstacles, the limited observing area of sensors, and controllable Field-of-View (FoV) of the Pan-Tilt-Zoom (PTZ) cameras. Besides, we reformulate the problem as a cooperative-competitive game and provide an interface to control the targets, i.e., the targets are controlled by adversaries to relentlessly challenge the camera policy with new strategies for the purpose of improving the robustness and generalizability of the trackers.

**Multi-Agent Learning Environments.** Besides the context of MCMT, we have seen other multi-agent learning (MAL) environments that service different tasks. Table 1 summarizes the most relevant MAL environments based on our literature review. To our findings, MATE stands out as the environment that simultaneously offers fully-cooperative & fully-competitive game types, Peer-to-Peer communication support, and heterogeneous agents.

**Self-Play and Population-Based Training Regime.** MATE experimented with three training principles to promote camera-target competition in zero-sum games. Solving zero-sum games can be highly non-trivial due to the non-transitivity (e.g., Rock-Paper-Scissor) in the policy space [65]. Conventional self-play makes the agent continuously play against the latest copy of itself. Since the agents in MATE are heterogeneous, we adopt the asymmetric version [66, 67, 68] of the self-play training method. However, self-play may fail to converge due to the lack of policy diversity [69, 70], thereby trapped by the non-transitivity. Fictitious Self-Play (FSP) [40] is a population-based method that maintains a policy memory storing past versions of the policy and uniformly samples a policy from memory as the response against the opponent. Policy Space Response Oracle [39] with Nash

Equilibrium solver (PSRO-Nash) is also a population-based method that computes a meta-strategy distribution. Instead of a uniform distribution, the distribution computed by PSRO-Nash resembles that of a mixed-strategy Nash Equilibrium. Recently, many efforts have been spent on extending PSRO methods to diverse PSRO methods [71, 72], no-regret PSRO methods [70], and PSRO with meta-learning [73, 74]. In this paper, we conduct experiments to demonstrate the effectiveness of these training regimes for improving and evaluating the robustness of the tracking agents.

# 3 MATE: the Multi-Agent Tracking Environment

In this section, we will introduce various details about the MATE environment. There are 4 kinds of entities in this 2D mini-world (shown in Fig. 1): $N_{\mathcal{C}}$ proactive cameras $\mathcal{C} = \{c_i\}_{i=1}^{N_C}$, $N_{\mathcal{T}}$ mobile targets $\mathcal{T} = \{t_i\}_{i=1}^{N_{\mathcal{T}}}$, $N_{\mathcal{O}}$ static obstacles, and $N_{\mathcal{W}}$ ($= 4$) warehouses storing cargoes. The reward structure inside MATE resembles the "min-max" nature of a cooperative-competitive multi-agent game. Camera agents must maximize their coverage rate collaboratively while minimizing repeated detection on the same target. In the meantime, targets transport cargoes between warehouses as fast as possible while minimizing the surveillance from the cameras. The role of obstacles in the environment is to provide temporary shelter against camera surveillance, but at the same time can act as roadblocks on the path to the destination for the target agents. The warehouses are scattered at the four corners of the mini-world.

Plenty of cargo needed to be delivered by the targets between the warehouses, in which the cargoes are priced based on the delivery duration.

## 3.1 Entities and States

We define the state as the internal attributes of the entities, which may change continuously as the environment progresses. Every agent (or controllable entity) may obtain its own states (public + private) but can only observe the public states of other agents.

**Camera** is an in-place, zoomable, directional sensor with a pie-shape field of view. The *publicly accessible state* of the camera $\boldsymbol{s}_c^{\text{pub}} = [x, y, r, R_s, \phi, \theta]$ contains the self-location data in the world coordinate system, the physical radius $r$, the visible line of sight $R_s$, the viewing direction angle $\phi$, and field-of-view angle $\theta$. In addition to these, the camera's *privately accessible state* $\boldsymbol{s}_c^{\text{pvt}} = \left[\boldsymbol{s}_c^{\text{pub}}, R_{s,\max}, \Delta\phi_{\max}, \Delta\theta_{\max}\right]$ are constants indicating the maximum possible values for these parameters, with $\Delta\phi_{\max}$ being the camera's maximum rotation speed and $\Delta\theta_{\max}$ being the maximum zooming speed.

**Target** is a mobile vehicle equipped with an advanced omnidirectional sensor for which obstacles would not block the sensing field. The *publicly accessible state* $\boldsymbol{s}_t^{\text{pub}} = [x, y, R_s, \mathbb{I}[\text{loaded}]]$ consists of the self-location data, the sensible range $R_s$, and an indicating variable $\mathbb{I}[\text{loaded}]$ that indicates whether the target is loaded with payloads. The *privately accessible state* $\boldsymbol{s}_t^{\text{pvt}} = \left[\boldsymbol{s}_t^{\text{pub}}, v_{\max}, W^{(1)}, \ldots, W^{(N_{\mathcal{W}})}, E^{(1)}, \ldots, E^{(N_{\mathcal{W}})}\right]$ are the maximum movement speed $v_{\max}$, a one-hot-like vector $\boldsymbol{W}$ to indicate the payload destination, and a bit array $\boldsymbol{E}$ to "memorize" if the previously visited warehouse is empty. There are two kinds of vehicles for targets, one with high speed and small carrying capacity, and the other with low speed and large carrying capacity. The former is twice as fast as the latter but has a halved carrying capacity.

**The Obstacle** is a circular-shape static object that randomly spawns (controlled by a distribution) in the environment. Targets may use obstacles to stay hidden from camera surveillance. We added a transmittance attribute to the obstacles, so the cameras with a particular chance can detect the target hidden behind the obstacle at each timestep. This design feature prevents the target agents from over-reliance on this shortsighted strategy. The state of an obstacle includes the location and the radius, i.e., $\boldsymbol{s}_o = \boldsymbol{s}_o^{\text{pub}} = [x, y, r]$.

## 3.2 Observations

Observation acquired by an agent is a partial representation of the true states of the environment. By default, the agents in MATE have partial observability over the environment. An agent could only observe the entities within its field of view and obtain the publicly accessible states of these observable entities.

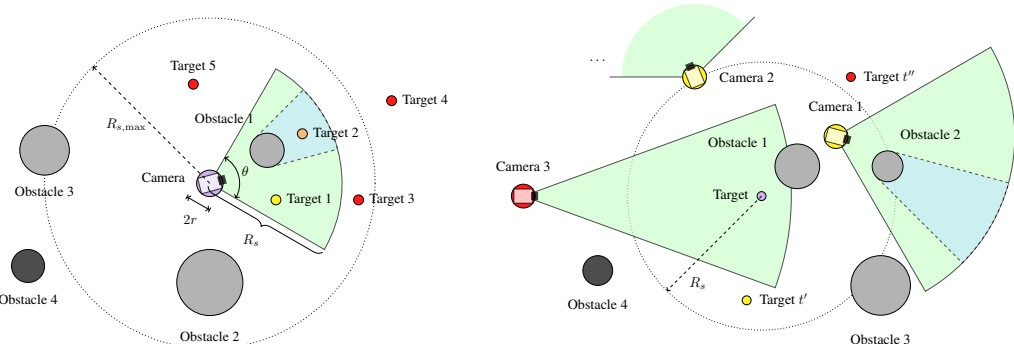

(a) Camera Observation (within the green sector)     (b) Target Observation (within the dotted circle)

Figure 2: Schematic diagram of agent sight ranges. The agent (at the center, colored in purple) can obtain its own privately accessible state, and other agents' and obstacles' publicly accessible states within the sight range (colored in yellow and gray). The camera can perceive the target (colored in orange) behind an obstacle with the probability value of the obstacle's transmittance.

**Camera observations** The observation of a camera $c \in \mathcal{C}$ can be divided into five parts, the preserved data $s^{\text{psrv}}$, its own privately accessible state $s_c^{\text{pvt}}$, and the publicly accessible states of targets and obstacles and other cameras within its field of view. The preserved data $s^{\text{psrv}}$ contains global information about the environment, such as the number of each category of entities, the current agent ID in the team, etc. Fig. 2a shows an example of the camera's perception in the green-shaded area.

**Target observations** The observation of the target $t \in \mathcal{T}$ is also composed of multiple parts, including the preserved data $s^{\text{psrv}}$, its own privately accessible state $s_t^{\text{pvt}}$; and the publicly accessible states of cameras, obstacles and other targets. Fig. 2b helps to demonstrate an example of the target's sight range.

Please refer to the supplementary for the mathematical formulation of the observations of both camera and target agents.

### 3.3   Actions

**Camera** is an in-place directional sensor with two types of continuous actions: rotation and zooming. These two action parameters can be adjusted simultaneously. Together they determine the shape of the camera's perception zone, though the area of perception remains unchanged. This design consideration balances the expected values of the number of perceivable targets under all possible action parameters.

**Target** is a mobile agent that can move freely inside this mini-world. Target's action space consists of the displacement vector $\boldsymbol{v} = (\Delta x, \Delta y)$ in Cartesian coordinates.

### 3.4   Reward Structure and Types of Stochastic Games

MATE can switch between three reward mechanisms corresponding to the typical three types of stochastic games. But first and foremost, we will motivate the use of Mean Coverage Rate as a reward function, which is an important metric used for evaluating MCMT tasks.

**Mean Coverage Rate** One typical method in RL for evaluating different models' performances on an equal basis is using the mean reward of every episode. However, we argue that mean coverage rate, while strongly correlating to episode mean reward, is comparatively a more intuitive measure for evaluating the performance of camera agents on tracking tasks. The mean coverage rate is also a normalized measure that eases the problem of comparing results from different environment configurations. For an episode with length $L$, it is:

$$\text{Mean Coverage Rate} = \frac{1}{L} \sum_{k=1}^{L} \frac{\text{Number of Detected Targets at step } k}{\text{Total Number of Targets}}. \tag{1}$$

In an intra-team *fully-cooperative game*, all agents in the same team will receive team rewards based on the team performance and would not differ between individuals. MATE, by default, uses Mean

Coverage Rate as the team reward for the camera agents. For the target agents, we propose the *transport reward* to incentivize the transport of goods with the awareness of avoiding detection. Formally, the target team reward is $r^{(\mathcal{T})} = F + B$ and empirically we keep $F$ and $B$ roughly equal. $F$ stands for "freight", a fixed-value sparse reward received upon every successful delivery of the assigned cargoes. $B$ is short for "bounty" on every cargo. The value of the bounty will depreciate per time-step and further decrease if the cameras have detected the target that carries this cargo. Whereas in the setting of inter-team *fully-competitive game*, we let the camera team receives the opposite value of the team reward of the targets, i.e., $r^{(\mathcal{C})} = -r^{(\mathcal{T})}$. The two teams are kept to play a zero-sum game in the environment. In addition to the built-in team rewards mentioned above, users can customize the reward functions with wrappers. Depending on the user configuration, the game setting may transform into a general *mixed-motive game*.

## 4 Core Features of MATE

**Sample Efficient and Easy to Use** MATE is a multi-agent environment based on numerical simulation and implemented in pure Python with minimal dependencies[1]. Users can install MATE with a single shell command. Without parallelization, a single-thread program can sample around 300 steps per second on a modern CPU[2] in the default configuration (4 cameras, 8 targets, 9 obstacles). Besides, MATE ships with various custom wrappers and built-in rule-based agents, and the existing algorithms can run on MATE with few modifications. The source code is released under the MIT Open Source License with detailed documentation. The MATE environment is out-of-the-box compatible with OpenAI Gym API [41]. We represent the sample code at the following that runs random action agents on our environment.

```python
import mate
env = mate.make('MultiAgentTracking-v0')  # or gym.make
env.reset()
done = False
while not done:
    camera_joint_action, target_joint_action = env.action_space.sample()
    (
        (camera_joint_observation, target_joint_observation),
        (camera_team_reward, target_team_reward),
        done,
        (camera_infos, target_infos)
    ) = env.step((camera_joint_action, target_joint_action))
```

**Communicative Agents** MATE implements an intra-team communication channel for each team that supports both broadcast and Peer-to-Peer communication. Unlike the widely used MPE environment [23], we explicitly isolate messages from agent observations so the user may customize the message format, such as vectors or texts. Communication facilitates strategic coordination among agents, preventing unnecessary exploration and repeated efforts, especially in a partially

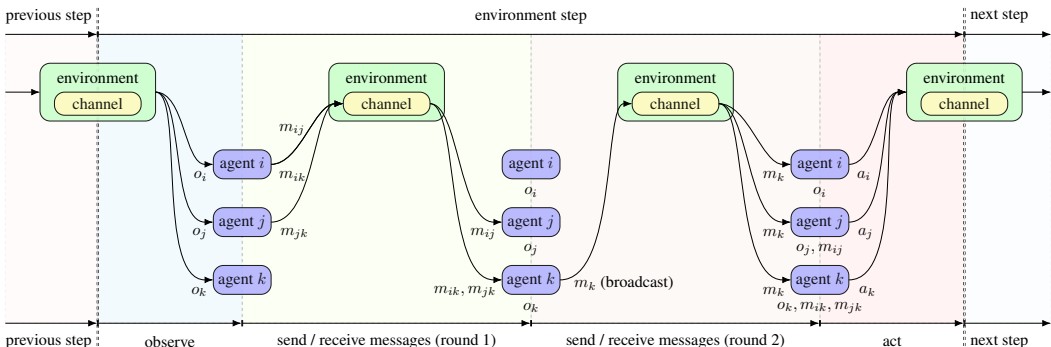

Figure 3: Illumination of the multi-round communication mechanism in MATE.

---

[1]Only NumPy, SciPy, Gym, and their dependencies are required.

[2]Tested using a single-threaded program with an Intel® i7 8700 @ 3.20GHz CPU. The agents take random action in the environment.

Table 2: Overview of the baseline algorithms. The TarMAC and I2C algorithms are communication add-ons that can utilize with other multi-agent reinforcement learning algorithms.

| Algorithm | Category | Centralized Training | On/Off-Policy | Action Space | Communication |
|---|---|---|---|---|---|
| QMIX [24] | Value-based | Yes | Off | Discrete | No |
| MADDPG [23] | Policy-based | Yes | Off | Continuous | No |
| IPPO [38] | Policy-based | No | On | Discrete / Continuous | No |
| MAPPO [25] | Policy-based | Yes | On | Discrete / Continuous | No |
| TarMAC [78] | Communication | Yes | On / Off | Discrete / Continuous | Broadcast |
| I2C [36] | Communication | Yes | On / Off | Discrete / Continuous | Peer-to-Peer |

observable environment. Users may add custom wrappers to simulate random signal noise, distance-based delays, restricted communication ranges, limited bandwidths, etc. As Fig. 3 illustrates, MATE supports multi-round communications, allowing agents to exchange several rounds of information within the same environment step. This mechanism can facilitate more multi-agent research regarding negotiations [75, 76] and conversations [77].

**Heterogeneous and Asymmetric**   The heterogeneity of the MATE environment exhibits two aspects: inter-team and intra-team. First, the agents in two opposite teams are completely distinct regarding their dynamics and tasks. Second, agents within the same team may have different abilities, e.g., the vehicles for targets with varying carrying capacities and movement speeds. These heterogeneities reflect a realistic theme and will allow agents to emerge with diversified strategies and individuality in a complex environment. The MATE environment hosts an asymmetric competitive game with two teams of heterogeneous agents. Asymmetry is not only reflected in the heterogeneity of agents but also in the variable quantities of players in the teams. Under different game setups, the equilibrium of the game change relative to the strength of both teams, and their corresponding strategy should also adapt accordingly.

**Variety, Flexibility, and Scalability**   In our default configuration, two groups of agents perform two tasks – the target coverage task for the camera agents and the transport task for the target agents. But with our user-friendly framework, researchers may extend this environment to suit more missions for the agents, such as deciding camera placement, trajectory prediction, resource collection, etc. The default setup of the environment reflects the most complex setting in which the following features are enabled: 1) *mixed cooperation-competition*, 2) *continuous action*, 3) *partially observable*, 4) *communicative*, and 5) *team reward only*. MATE is highly modularized so that the users can convert to different environments to suit their particular needs with our provided wrappers (presented in the supplementary). Users can train a target or camera network curriculum by dynamically adjusting the difficulty levels and transferring these policies across different settings. The number of entities in the environment is also configurable. MATE may support simultaneous interactions between two to hundreds of agents. As the quantity of agents varies, the complexity and difficulty of the environment also change accordingly, which allows the emergence of diverse strategies. Simply by varying the number of agents on both sides, the users can test the robustness of the newly-developed Multi-Agent Learning (MAL) algorithms.

## 5   Experiments

In this section, we will present the results for **(1)** collaborative game where training one team of agents (either cameras or targets) against rule-based opponents **(2)** additionally incorporates multi-agent communications into the collaboration games **(3)** competitive game where training two teams of agents using asymmetric self-play or Population Based Training (PBT).

For fair comparisons in the cooperative games, we ran each experiment between different algorithms for 10 million environment steps. The model performance was averaged across experiments from three random seeds to reflect statistical properties. We use and extend the RLlib [42] to implement QMIX [24], MADDPG [23], IPPO [38], MAPPO [25], TarMAC [78], and I2C [36] algorithms in all of our experiments. Table 2 lists the properties of the baseline algorithms.

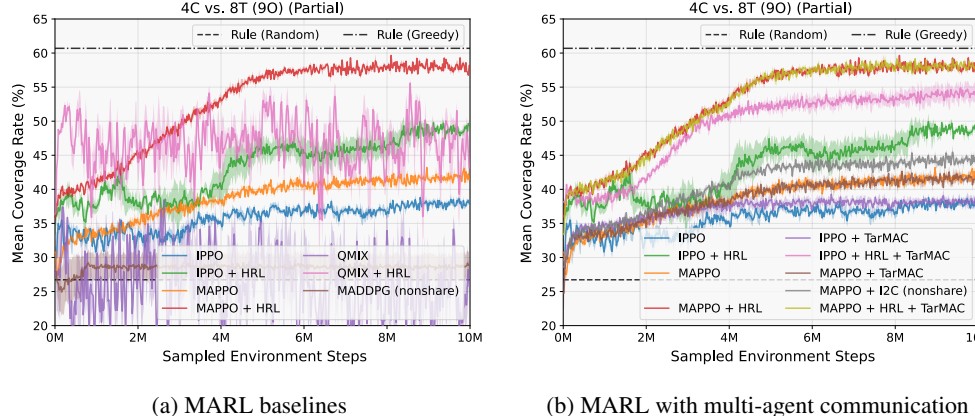

(a) MARL baselines        (b) MARL with multi-agent communication

Figure 4: The learning curves for the camera agents in the cooperative game. The intervals (shaded region) indicate one standard deviation over three separate runs. The target agents are (greedy) rule-based controlled.

## 5.1 Training Cameras in Fully-cooperative Game

We conduct feasibility checks on the train-ability of camera agents using MARL algorithms on MATE and accordingly report the training performance of the camera's policies on the 4C vs. 8T (9O).[3] setting. We presented a hierarchical RL (HRL) model with a selection-based low-level policy, which is controlled by a rule-based executor. The MARL algorithms only learn the high-level policy. Moreover, we added a 5-steps frame-skip to the high-level policy, in which the low-level policy would receive a signal from the high-level policy once in 5 timesteps.

The results in Fig. 4a demonstrate the superior performance of the HRL methods and the steady convergence of the PPO-based methods. Fig. 4b shows the effect of adding multi-agent communication modules to various MARL algorithms on the Mean Coverage Rate. Enabling communication for the MAPPO algorithm with the hierarchical agent structure hardly shows improvement in convergence. We believe it is because of the strong inductive bias of the HRL method, given that the policies for the low-level executors are based on pre-set rules (recall that only the manager policy is trained).

## 5.2 Training Targets in Fully-cooperative Game

In this section, we run multiple MARL algorithms to train the target agents competing against rule-based (greedy) cameras. As expressed in Fig. 5, target agents experience more conclusive performance gains by incorporating the multi-agent communication module compared to the camera agents shown in Fig. 4b. IPPO with the TarMAC protocol may achieve comparable performance to MAPPO+TarMAC, given that the critic model of the latter additionally has access to the global states. MAPPO+I2C attains the best convergence out of all methods implemented. Fig. 6 reflects the difficulty in normalized reward for two different settings. In 2C vs. 4T (0O), partial observability does not significantly hinder learning for MAPPO but causes an approximately 0.35 decrease in normalized episode reward for the IPPO. In 4C vs. 8T (0O), IPPO failed

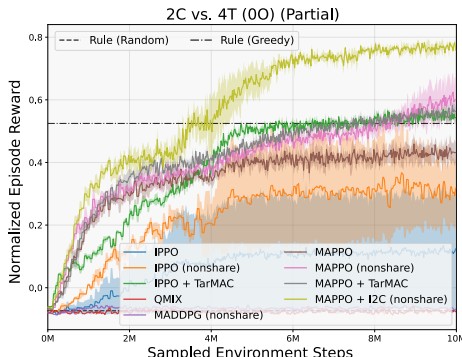

Figure 5: The learning curves (with intervals of one standard deviation across three separate runs) of various MARL methods in training target agents. Rule-based greedy agents control the cameras.

---

[3]Abbreviated "camera" as "C", "target" as "T" and "obstacle" as "O". Collectively 4C vs. 8T (9O) refers to a game setting where four camera agents play against eight target agents with nine obstacles in the environment

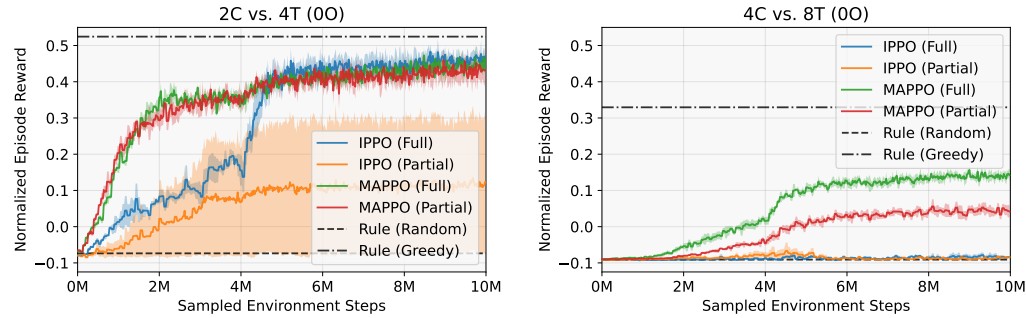

Figure 6: A comparison between two different settings shows the significance of the observability mode in learning meaningful policies for the target agents.

in both modes of observability while MAPPO suffered an approximately 0.1 decrease in the reward when switched to partial observability.

## 5.3 Zero-sum Fully-competitive Game

In Section 5.1 and 5.2, we present the results of training camera or target agents against fixed-policy opponents. However as these models implicitly treat their opponents as integrated parts of the non-stationary environment, this would often result in over-fitting or failure to generalize against new opponents [39]. In realistic deployment, a stable and robust solution is often preferred over a better-performing but brittle solution. Therefore to improve the robustness of the camera policy, we proposed to train camera and target agents in co-evolution with a zero-sum payoff structure. For the experiment, we trained the agents with three population-based methods: PSRO-Nash [39], Fictitious Self-Play [40], (asymmetric) self-play, and present the performance comparison in Fig. 7.

Note that the exploitability of a policy or a population of policies is an intuitive measure of robustness. The exploitability, in the context of MATE, is formally defined as:

$$\text{exploitablility}(\Pi^{\mathcal{C}}, \Pi^{\mathcal{T}}) = \frac{1}{2} \sum_{i \in \{\mathcal{C}, \mathcal{T}\}} \left[ U^i\big(\mathbf{BR}(\Pi^{-i}), \Pi^{-i}\big) - U^i\big(\Pi^i, \Pi^{-i}\big) \right], \tag{2}$$

where $\mathcal{C}, \mathcal{T}$ refer to "camera group" and "target group", $-i = \{\mathcal{C}, \mathcal{T}\} \setminus \{i\}$, $\mathbf{BR}$ stands for "best response", $U^i(\cdot, \cdot)$ ($i \in \{\mathcal{C}, \mathcal{T}\}$) are the utility functions, and $\Pi^i$ ($i \in \{\mathcal{C}, \mathcal{T}\}$) are the policy populations accordingly. For fully-competitive settings, $U^{\mathcal{C}} + U^{\mathcal{T}} = 0$.

The exploitability can be interpreted as the average performance difference between the best-response (BR) and current policies. Low exploitability implies that both opposing groups have approximately converged to their best-response policies at the current iteration, indicating proximity to the Nash equilibrium. In Fig. 7, both sub-figures show that PSRO-Nash and self-play can converge to policy populations that are less exploitable than the populations trained against non-

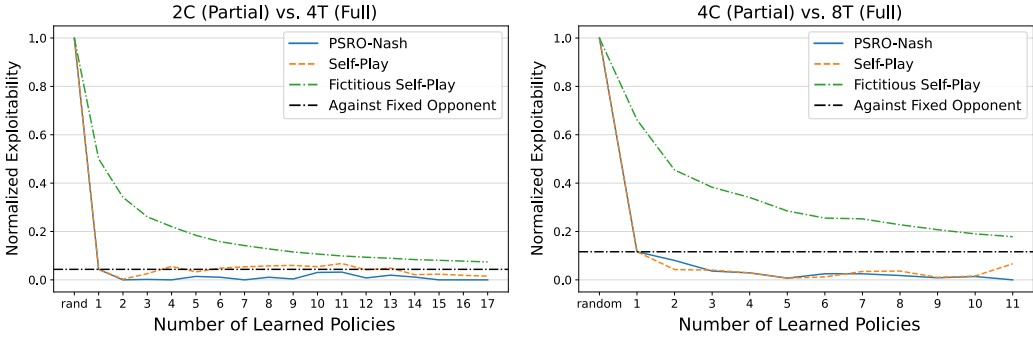

Figure 7: The exploitability of both the populations of cameras' and targets' policies at each population iteration. The left and right figures differ in the settings of the environments.

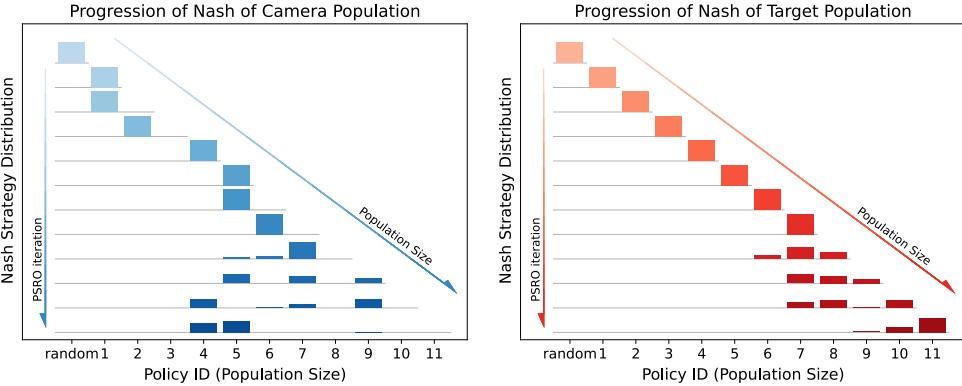

Figure 8: The probability distribution of the meta-strategy for each PSRO iteration for the policy population of both camera and target teams. Both teams adaptively change their policy according to the adversaries. The agents are trained under the 4C vs. 8T (00) configuration.

evolving, fixed-policy opponents. Since Fictitious Self-Play uniformly samples policy from the policy distribution, the random policy has an equal probability of being sampled as the other policies in the distribution at every population iteration, which therefore explains the slow decreasing trend in exploitability. Fig. 8 visualizes the progression of distribution for the meta-strategy taken by the players (the camera team and the target team). A relatively scattered Nash distribution indicates the presence of multiple plausible strategies.

## 6  Conclusion

Multi-Agent Tracking Environment (MATE) is a novel multi-agent simulation for benchmark multi-agent reinforcement learning algorithms in the target coverage problem. MATE hosts a zero-sum cooperative-competitive, asymmetric game between "cameras" and "targets", in which cameras gain team rewards to maximize the number of the covered targets. MATE incorporates the intra-team peer-to-peer (P2P) communication feature and trainable adversarial target agents. This environment is built purely in Python and integrated with OpenAI Gym API [41] enabling great compatibility and extensibility to most distributive RL frameworks. The lightweight of this environment also ensures high sampling efficiency. MATE allows for flexible configurations for the simulation environments along with highly customized scenarios to fulfill specific research needs. We conduct a series of benchmarks to show the performance of target and camera agents trained by MARL methods and algorithms in various settings. We hope this work will serve as a useful guide for the community using MATE to conduct further research.

**Limitations.** The first-stage focus of MATE is to provide an all-in-one benchmark for testing various MARL algorithms and a new platform for studying distributed target coverage tasks with trainable adversaries. Admittedly, the focus of MATE is lesser on the aspects of visual perceptions, for example, evaluations in three-dimensional space. For these purposes, one of our future works will extend MATE into the high-quality 3D game engine, e.g., developing realistic environments on Unreal Engine 4 (UE4) with UnrealCV [79].

**Fair Use of the Dataset.** Despite MATE was not purposefully designed for scenarios that incur direct violation of privacy or harm the well-being of others, the theme of MATE brings up a valuable topic of discussion with regard to a recognized conflict between the advancement of AI technologies and the integrity of social well-being. Although the possibility of direct application of MATE to other exploitative systems remains slim, the transfer of knowledge between multi-agent tracking systems is theoretically plausible. While enjoying the benefits of training smarter camera systems using MATE, we do have genuine concerns about the negative societal impacts due to the misuse of tracking technologies in repressive surveillance. We hereby advocate for more responsible use of multi-agent tracking environments including MATE and we condemn the act of using MATE or other similar systems for malicious activities.

# 7 Acknowledgements

This project was supported by MOST-2018AAA0102004, NSFC-62061136001, China National Post-doctoral Program for Innovative Talents (Grant No. BX2021008), and Qualcomm University Research Grant. We also thank Jing Xu, Yurong Chen, and Yuanfei Wang for their insightful discussions.

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
