# OpenReview forum: "MATE: Benchmarking Multi-Agent Reinforcement Learning in Distributed Target Coverage Control"
_NeurIPS.cc/2022/Track/Datasets_and_Benchmarks — NeurIPS 2022 Datasets and Benchmarks _

### Official Review · Reviewer_svSJ · 2022-06-30
**I can see authors' ambitions in this new MARL environment but this paper needs more novelty**

**Rating:** 7
**Confidence:** 4
**Correctness:** 1. Line 196
**Clarity:** Yes, this paper is well written and w…

**Strengths:**

1. In my opinion, this is the first open-sourced and standardized multi-agent environment specifically for target coverage problem. It provides detailed docs and clear coding tutorials for target coverage problem-related researchers. It is integrated with the OpenAI Gym API and can be used with RLib, Tianshou, Stable-Baselines-3, etc as the authors claimed.

2. Dislike other multi-agent environments using broadcast communication regime (which is naïve), MATE provides peer-to-peer communication options. I believe such features provide convenience and some inspires for studying communication problems in multi-agent system.



**Weaknesses:**

1. Though this work make up the gap that there is no standard environment for the target coverage control problems, this paper does not clarify the importance in the field of target coverage control problems. The references about the target coverage control problems are supposed to be added.

2. Though Some features of META are highlighted such as the Game Type, Communication Method, Agent Type and the Scalability, I did not see the superiority of META compared to other multi-agent environments. META is specifically designed for the target coverage control problem. All these highlighted features are from the problem setting not from the META itself.

3. META is more like an extension work of this paper: "Learning Multi-Agent Coordination for Enhancing Target Coverage in Directional Sensor Networks" which provided some environments resource. Do you think is it an incremental work?

4. The paper "Learning Multi-Agent Coordination for Enhancing Target Coverage in Directional Sensor Networks" has a good method to solve the problem. The HiT-MAC method should be performed as well in the experiments. If the HRL model is some method similar to Hit-MAC, the author should cite the paper and explain the difference and similarity between them.

**Additional Feedback:**

The novelty is not obvious, but I am happy to see this high-quality engineering work contributing to our research society. To make this benchmark paper more attractive,  I think the authors should more focus on the target coverage problem itself instead of enthusiastically promoting its general comparison with other multi-agent environments.

**Documentation:**

Authors provides high quality documentations.

**Ethics:**

No.

**Relation To Prior Work:**

No.

It did not include the method in "Learning Multi-Agent Coordination for Enhancing Target Coverage in Directional Sensor Networks" as a baseline. It neither discusses the difference/improvement compared to that paper.

MATE has some similarities to the multi-agent environment PuckWorld and WaterWorld. But it did not mention them. There are a lot of multi-agent environments, the authors should better show MATE's novelty or show some new marl algorithms in the paper.

**Summary And Contributions:**

This paper introduces a multi-agent tracking environment (MATE) where a  group of camera agents trying to cover target agents as much as possible, while another group of target agents try to escape from the coverage of camera agents. Then the authors perform several MARL algorithms (MAPPO, IPPO, QMIX and MADDPG) for cooperative tasks in MATE. Different communication protocols (TarMAC, I2C) are also tested in cooperative tasks. Some other experiments such as effectiveness of fictitious play in competitive tasks is investigated. This new MARL environment is provided with high-quality docs.

---

> ### Author Response · Authors · 2022-08-22
> **Response to Reviewer svSJ (5/5)**
>
> > **AF13:**  The novelty is not obvious, but I am happy to see this high-quality engineering work contributing to our research society. To make this benchmark paper more attractive, I think the authors should more focus on the target coverage problem itself instead of enthusiastically promoting its general comparison with other multi-agent environments.
>
> **A13:**
> Thanks for your appreciation and the suggestions. However, we genuinely believe that MATE contributes to both sub-fields mentioned above of MARL.
>
> Regarding the contribution of MATE as a new multi-agent learning environment, we present a new benchmark that concerns several prominent research topics in the study of multi-agent learning, such as multi-agent communication/negotiation, asymmetric game, multi-agent credit assignment, scalability, opponent modeling, and decentralized coordination. These research points exist in the target coverage control problem and similarly exist in wide range of real-world applications. Thus, we argue that benchmarking the off-the-shelf MAL algorithms in MATE can expose and project possible weaknesses in deployment and push the community toward studying more practical algorithms for real-world applications.
>
> Regarding the contribution of MATE as a novel environment for studying the target coverage control problem, it provides an easy-to-use simulation for training the learning-based methods and evaluating the multi-agent tracking system on coverage rate and communication efficiency. The procedural generator helps the users generate new environments with different configurations. MATE has environmental features (e.g., procedural generation of scenarios, random obstacles) and task features (e.g., transport task, bounty reward) to encourage dynamic competition between the two rival groups and to allow the emergence of more diverse and interesting strategies. The controllable targets can enable the adversarial training framework to build more robust control policies for the tracking systems.

---

> > ### Comment · Reviewer_svSJ · 2022-08-24
> > **Thank the authors for these replies**
> >
> > Thank you so much for replying to my questions and resolving my concerns. I can see it takes a lot of time and effort. Though I and the authors may have a different standard for a database and benchmark paper,  I am convinced that this paper, this MATE tool is qualified for publication and is deserved for making more people use it. I am looking forward to use it in my future research work.

---

> > > ### Author Response · Authors · 2022-08-24
> > > **Thanks again for your time and consideration**
> > >
> > > We are glad to see that we have addressed your concerns, and we are deeply grateful for your appreciation. Should you have any concerns regarding the usage of the environment, please feel free to post an issue on GitHub or shoot us an email :)

---

> ### Author Response · Authors · 2022-08-22
> **Response to Reviewer svSJ (4/5)**
>
> Continue to the previous response:
>
> We perform extra experiments to compare the HiT-MAC with MAPPO+HRL. Because MATE is quite different from the environment in the original HiT-MAC paper, we changed some properties of the HiT-MAC algorithm to fit the partial observability in MATE. Such as, only the targets in the sector-shaped area can be observed, and the lower-level policy can only select to track the observable targets. The training curves are shown in Fig. H.1 (supplementary) and [Exhibit E](https://sites.google.com/view/mate-neurips2022/home). Both HiT-MAC and MAPPO+HRL converge to similar performance. The HiT-MAC algorithm results in slower convergence because its high-level policy is a centralized agent. It has to model the joint policy with higher dimensions of observation and action space ($2^{nm}$), which results in a harder exploration when there are many targets.
>
> ------
>
> > **RTPW11:** MATE has some similarities to the multi-agent environment PuckWorld and WaterWorld. But it did not mention them.
>
> **A11:**
> Thanks for your reminder. Although WaterWorld and PuckWorld resemble similarities in artistic style to MATE, this is done so to conserve computational efficiency. But MATE and these two Pygame learning environments differ in complexity and customizability. MATE is a gamification of the MCMT tracking task, in which both the cameras and the targets are trainable and should learn to exploit their opponent’s weaknesses. The objectives in MATE for either party also have a min-max nature (e.g, maximize transport rate while avoiding detection, track most targets in local view while ensuring maximum team coverage), while in PuckWorld and WaterWorld the objectives are straightforward. In terms of customizability, MATE has a procedural generator for maps and a lot of customizable properties.
>
> In addition, [PuckWorld](https://pygame-learning-environment.readthedocs.io/en/latest/user/games/puckworld.html) and [WaterWorld](https://pygame-learning-environment.readthedocs.io/en/latest/user/games/waterworld.html) environments are **single-agent environments** and therefore out-of-scope to our study of related work. The multi-agent version of [Waterworld](https://github.com/Exception4U/MultiAgent-Waterworld) is written in JavaScript, which is susceptible to compatibility issues when the training scripts and deep-RL framework are written primarily in Python.
>
> ------
>
> > **RTPW12:** There are a lot of multi-agent environments, the authors should better show MATE's novelty or show some new marl algorithms in the paper.
>
> **A12:**
> We have listed a detailed comparison between relevant multi-agent environments in the related work section. The key novelty is that MATE is an **all-in-one** multi-agent environment and gamification of the target coverage control problem.
> Unlike most gamified environments, MATE is an abstraction of the real-world scenarios that account for the critical elements needed for a successful real-world multi-agent application. These include scalability, robustness, and communication efficiency. MATE provides interfaces for peer-to-peer communication and training adversaries with customizable capabilities (speed, geometry, capacity, etc.) for these purposes. We also provide an easy-to-use interface to customize the environments and generate various configurations using our procedural generator for research purposes. For example, (1) the users can change the number of agents and targets for evaluating the scalability of the proposed solution, (2) train a population curriculum, (3) choose the communication modes for analyzing the communication efficiency, and (4) generate diverse layouts for evaluating the robustness of a multi-agent solution.
> MATE also has additional design features to enhance the gamification of the MCMT task. There are randomly scattered obstacles in the environment where targets can use them to their advantage against the cameras, and targets are forced to perform transport tasks. As described, the environment is carefully designed to prevent one-sided, dominant strategies (e.g., targets hiding behind obstacles permanently, or targets without obstacles would easily be exploited by the cameras, etc.).
>
> Also, we would like to kindly remind the reviewer that formulating new MARL algorithms is outside of the scope of this paper and potentially out of scope for the Dataset and Benchmark Track.

---

> ### Author Response · Authors · 2022-08-22
> **Response to Reviewer svSJ (3/5)**
>
> > **W3:** META is more like an extension work of this paper: "Learning Multi-Agent Coordination for Enhancing Target Coverage in Directional Sensor Networks" which provided some environments resource. Do you think is it an incremental work?
>
> > **W4:** The paper "Learning Multi-Agent Coordination for Enhancing Target Coverage in Directional Sensor Networks" has a good method to solve the problem. The HiT-MAC method should be performed as well in the experiments. If the HRL model is some method similar to Hit-MAC, the author should cite the paper and explain the difference and similarity between them.
>
> > **RTPW10:** It did not include the method in "Learning Multi-Agent Coordination for Enhancing Target Coverage in Directional Sensor Networks" as a baseline. It neither discusses the difference/improvement compared to that paper.
>
> The reviewer raised multiple concerns regarding a comparison between the target coverage environment proposed in HiT-MAC and MATE:
>
> > - MATE could be an incremental work of the paper "Learning Multi-Agent Coordination for Enhancing Target Coverage in Directional Sensor Networks" (HiT-MAC for short)
> > - Similarity and difference between HiT-MAC and MATE, in terms of the design of the environment
> > - A comparison between the HRL methods used in MATE and HiT-MAC
> > - Add HiT-MAC as a baseline
>
> **A3 & A4 & A10:**
> We do not think MATE is incremental work to HiT-MAC. Even though these two works both try to bridge the gap between the target coverage problem and the multi-agent learning,  they differ from many perspectives.
>
> * First, the two works are of different focuses. HiT-MAC aims to develop a hierarchical multi-agent coordination framework for the directional sensor networks. In contrast, MATE aims to provide an all-in-one environment to benchmark the multi-agent learning algorithms in the target coverage problem.
> * Second, the environments provided in MATE are much more general (as in flexible configuration) and complex than HiT-MAC. The environment provided in HiT-MAC can be regarded as a significant simplification of MATE due to the following aspects: (1) sensors can not adjust the FoV, (2) sensors have full observability over all the target states at all times, and (3) there are no obstacles.
> * Third, we make the targets trainable, enabling the co-evolution mechanism to help improve the generalization and robustness of the camera policy. In HiT-MAC, the targets are only controlled by a random walking policy. This would often result in over-fitting or failure to generalize against new targets. In Fig. 7, both sub-figures show that PSRO-Nash and self-play can converge to policy populations that are less exploitable than the populations trained against non-evolving, fixed-policy opponents.
> * Fourth, we provide two communication mechanisms (peer-to-peer and broadcast) in MATE. With that, we can explore and benchmark different methods (from centralized to decentralized) in our environment. Also, the peer-to-peer communication feature will be helpful for research in emergent language and emergent communications.
>
> We list the properties of HiT-MAC and MATE in the tables below:
>
> | Algorithm | High Level | Low Level | Centralized Training | Decentralized Execution | Communication |
> | :--- | :--- | :--- | :--- | :--- | :--- |
> | HiT-MAC | a superagent jointly making decision | learned or rule-based | Yes | No  | all-to-one & one-to-all |
> | HRL in MATE | decentralized | learned or rule-based | Yes | Yes | none or learned (broadcast or peer-to-peer) |
>
> | Environment | Game Type | Observation | Action | Agent Type |
> | :--- | :--- | :--- | :--- | :--- |
> | HiT-MAC | fully-cooperative | targets in the circle area can be observed, even when they are not covered by the camera, obstacles are not considered | only rotation with fixed viewing-angle | homogeneous  |
> | MATE | fully-cooperative & fully-competitive | only targets in the sector-shaped area can be observed, obstacles will occlude the camera's sight | rotation and zooming | heterogenous |
>
> | | Contribution |
> | :--- | :--- |
> | HiT-MAC | develop a semi-centralized hierarchical multi-agent coordination framework for the directional sensor networks |
> | MATE    | a new multi-agent learning environment, which supports both fully-cooperative & fully-competitive game types, multi-agent communication, various settings & curriculum learning, and heterogenous agents |

---

> ### Author Response · Authors · 2022-08-22
> **Response to Reviewer svSJ (2/5)**
>
> > **C7:** Line 246: "We use and customize the RLlib implementations of QMIX, MADDPG, PPO, TarMAC and I2C algorithms in all of our experiments."
> > I do not understand how the PPO of RLlib can be naturally extended to MAPPO.
>
> **A7:**
> The MAPPO algorithm is a multi-agent extension of PPO. In the MAPPO algorithm, the actor takes the local observation to produce the action, while the critic takes the global state as input and outputs the value estimation. We implement a gym wrapper at [wrapper `RLlibMultiAgentCentralizedTraining`](https://github.com/UnrealTracking/mate/blob/aaecc56793a70cc06d4a2bcc5f39d45fd6b674c5/examples/utils/wrappers.py#L131-L141) using a `gym.spaces.Dict` space as the observation space.
>
> ```python
> observation_space = spaces.Dict(
>     spaces = OrderedDict(
>         [
>             # Local observation of the current agent
>             ('obs', env.observation_space),
>             # Global state of the environment
>             ('state', env.state_space),
>             # Joint action for other agents (exclude the current agent)
>             ('prev_others_joint_action', env.others_joint_action_space),
>         ]
>     )
> )
> ```
>
> Then in the policy model, the actor network only takes the local observation `'obs'` as input, and the critic network takes the global state as the input. The source code of our MAPPO model can be found at [here](https://github.com/UnrealTracking/mate/blob/aaecc56793a70cc06d4a2bcc5f39d45fd6b674c5/examples/mappo/models.py#L107-L128). We also use a policy mapping function to share the policy parameters for all the agents in the same team. Which maps all agents to the same policy ID ([source](https://github.com/UnrealTracking/mate/blob/aaecc56793a70cc06d4a2bcc5f39d45fd6b674c5/examples/utils/rllib_policy.py#L75-L76) and [source](https://github.com/UnrealTracking/mate/blob/main/examples/mappo/camera/config.py#L98-L103)). In evaluation, only the actor network and the local observation is used.
>
> ------
>
> > **C8:** I am wondering why HRL was not applied to training targets in a fully-cooperative game?
>
> **A8:**
> We found that the MARL algorithms can learn good strategies for the target agents without utilizing HRL. In fact, we have already developed an HRL algorithm for controlling the target agents, in which the higher-level policy learns to decide subgoals (waypoints), and the lower-level policy is a rule-based executor performing navigation to these assigned subgoals. We found that the performance of the training-from-scratch MARL model is on par with the target-agent HRL method.
>
> ------
>
> > **C9:** I do not understand the necessity of subsection 5.3 in Experiments. Did these experiments show any novelty of META?
>
> **A9:**
> Training camera agents in autocurricula is the main motivation for developing this environment. Our novelty does not lie in proposing new algorithms but in developing a novel environment to enable training cameras with population-based training regimes or self-play methods.
>
> We have added more descriptions regarding population-based training regimes and the self-play method in both Section 5.3 and the related work section.

---

> ### Author Response · Authors · 2022-08-22
> **Response to Reviewer svSJ (1/5)**
>
> Thank for your insightful comments. Please kindly refer to the [Common Response](https://openreview.net/forum?id=SyoUVEyzJbE&noteId=GbmiOtZi_G0) at above for the list of changes in our manuscript and supplementary material, and a dedicated website ([link](https://sites.google.com/view/mate-neurips2022)) for our rebuttal. We look forward to further discussion.
>
> ------
>
> > **W1:** Though this work make up the gap that there is no standard environment for the target coverage control problems, this paper does not clarify the importance in the field of target coverage control problems. The references about the target coverage control problems are supposed to be added.
>
> **A1:**
> We believe that we have listed a couple of examples in the first paragraph of the introduction. These real-life applications and commercial projects that bring societal benefits, such as anti-poaching, anti-smuggling, border security, and ball-tracking, all motivate the importance of studying the target coverage control problem.
>
> We apologize for the mismatch between our introduction and the related work section. Most of the related works concern “Multi-Camera-Multi-Target” tracking, which in our opinion, is a sub-field of the target coverage control problem. Hence, we have clarified further in the related work section of the updated manuscript.
>
> ------
>
> > **W2:** Though Some features of META are highlighted such as the Game Type, Communication Method, Agent Type and the Scalability, I did not see the superiority of META compared to other multi-agent environments. META is specifically designed for the target coverage control problem. All these highlighted features are from the problem setting not from the META itself.
>
> **A2:**
> Though the main theme of MATE concerns the target coverage control problem, MATE itself is not fixated on solving this problem. The user can simulate other scenarios that resemble some real-life uses with simple modifications. For example, by removing all cameras in the configuration file and randomizing the location of the warehouses, we can turn this environment into a testbed for heterogenous multi-agent task planning or trajectory planning for the target agents. This modified environment can be used for research in navigation, object searching, and object transporting. We have showcased a possibility in Figure 1e, it demonstrates a pure navigation scenario. All of these benefits can attribute to the easy customizability, extendability, and user-friendliness of MATE. Moreover, we surveyed all the existing MARL environments and we reported the study in Table 1, which highlights the difference in key features between MATE and other MARL environments.
>
> ------
>
> > **C5:** Line 196: "MATE is a multi-agent environment based on numerical simulation and implemented in pure Python with minimal dependencies."
> > If you claim "minimal", you should provide a table or some references to verify it. And the interpretations of "dependencies" can be varied. This claim seems incorrect to me unless you add some more constraints.
>
> **A5:**
> By minimal dependencies, we meant the minimal pip dependencies. This ensures a simple installation process and is easier to manage. We have added a footnote in our manuscript for clarification: “Only NumPy, SciPy, Gym, and their dependencies are required”. Because MATE relies mostly on Python packages, it also has good cross-platform (Windows, macOS, Linux) support.
>
> ------
>
> > **C6:** Line 238: "newly-developed MAL algorithms"
> > What is MAL algorithm?
>
> **A6:**
> MAL stands for multi-agent learning algorithms. We wrote MAL algorithms because it covers more algorithms that MATE can potentially support. MAL covers MARL, regret-based learning, self-play, evolutionary algorithms, etc. [1],[2].
>
> [1] Panait, L., & Luke, S. (2005). Cooperative Multi-Agent Learning: The State of the Art. Autonomous Agents and Multi-Agent Systems, 3(11), 387-434.
> [2] DiGiovanni, A., & Zell, E. C. (2021). Survey of Self-Play in Reinforcement Learning. arXiv preprint arXiv:2107.02850.

---

### Official Review · Reviewer_7uYR · 2022-07-17
**A useful benchmark for mixed cooperative-competitive multi-agent reinforcement learning**

**Rating:** 7
**Confidence:** 4

**Strengths:**

* This is an interesting new benchmark for mixed cooperative-competitive multi-agent reinforcement learning with sufficient customizability options and ease of use due to its OpenAI Gym interface. The problem it models also seems relevant and I am not aware of another such benchmark for multi-agent tracking and coverage.
* There are various built-in opponent strategies for quick testing as well as multiple reward structures
* Various communication options are available to agents, thereby enabling full communication settings as well as restricted communication settings
* The agents in a team can be heterogeneous and the teams can play asymmetric games
* The performance of various popular multi-agent reinforcement learning solutions are benchmarked against the proposed environment, thereby providing a nice benchmarking in the context of multi-agent coverage and tracking

**Weaknesses:**

* I do not see any substantial weaknesses outside of the writing, which can be improved.

**Additional Feedback:**

N/A

**Clarity:**

The writing can use some work. While the paper is comprehensible, the writing is clumsy enough to hinder the flow of the paper. Some examples, though there are many more:
* "fully-competition" -> fully-competitive
* "gained many interests" -> gained much interest
* "The competitive game hosted in MATE involve two-team of heterogeneous agents, More importantly" -> ...involves two teams of heterogeneous agents. More importantly

**Correctness:**

The results seem correct and the reward metrics being adopted are reasonable for the fully cooperative and mixed cooperative-competitive settings in multi-agent reinforcement learning.

**Documentation:**

There is sufficient detail in the documentation.

**Ethics:**

Though the authors cite a specific application domain as being potentially sensitive, I do not think this is a concern since this is a very high-level abstraction that can apply to various domains.

**Relation To Prior Work:**

Prior work is properly referenced and put in the context of this work in Table 1.

**Summary And Contributions:**

The authors propose a new OpenAI Gym environment for the partially observable two-team target coverage problem. The environment comes with wrappers for built-in rule-based strategies to test against, three types of reward signals, and various configuration options for the number of agents in each team, number of obstacles, radii parameters, etc. The authors also benchmark various existing multi-agent reinforcement learning solutions in this environment for the fully cooperative settings where the opposing team strategy is fixed as well as the mixed cooperative-competitive setting where both agents are learning.

---

> ### Author Response · Authors · 2022-08-22
> **Response to Reviewer 7uYR**
>
> Thank for your insightful comments. Please kindly refer to the [Common Response](https://openreview.net/forum?id=SyoUVEyzJbE&noteId=GbmiOtZi_G0) at above for the list of changes in our manuscript and supplementary material, and a dedicated website ([link](https://sites.google.com/view/mate-neurips2022)) for our rebuttal. We look forward to further discussion.
>
> ------
>
> We thank Reviewer 7uYR for your positive feedback. We have fixed the clarity issues you kindly pointed out, and we apologize for any inconvenience caused. We made our best effort to improve the quality of writing of our manuscript and revised the manuscript thoroughly. Should you have any further questions, please let us know.

---

### Official Review · Reviewer_4Kzr · 2022-07-27
**Well-written paper, interesting environment but not enough valid experiments for training.**

**Rating:** 5
**Confidence:** 3

**Strengths:**

1. It is not original to track some targets because there are environments for tracking active objects [1]. In some ways, however, it appears novel. Tracking multiple agents (targets) with multiple agents (cameras) initially appears novel. Second, it appears novel to combine tracking tasks with cargo shipments, which enables a zero-sum game.
2. I believe the MATE environment has reached completion.
3. As soon as I saw the environment, I found it intriguing that it could serve as a basic simulator for real-world applications. It could provide a guideline for algorithm and application researchers in the field of tracking and cargoes that are not exposed to anything.

**Weaknesses:**

1. The GUI closely resembles Multi-Agent Particle Environment (MPE), which contradicts the authors' claims of realism. Therefore, it is insufficient for authors to advocate realistic characteristics.
2. Baselines are not enough to show the validity of the environment, as shown in Fig 5. Lines are so flat that it doesn't looks like being under training for 10 million steps.
3. So short Related Work section that I felt hard to understand in some sections like 5.3.

**Additional Feedback:**

1. What effect does 'minimizing the exposure time to cameras' have on target agents in terms of real-world applications?
2. Env SMAC is written to be non-scalable in table 1, but I believe it is scalable because it can produce more units depending on the settings.
3. In Figure 3, I wonder how to choose between peer-to-peer and broadcast methods for passing messages. How do agents choose a peer when using the peer-to-peer method?
4. Do you have any experience with curriculum learning for training the algorithms in Figure 5 that were not trained?
5. In section 5, the HRL model receives a high-level signal from the MARL algorithm and selects a rule-based action. However, since the signal may be a latent vector, isn't it necessary to train the HRL model with the signal as an input?
6. I want to know the selection criteria for baseline algorithms (IPPO, MAPPO, QMIX, MADDPG). There are SOTA multi-agent algorithms that are policy gradient and value-based in cooperative and competitive environments.
7. I have a strong suspicion regarding the training camera experiments in the fully cooperative game. Despite 10 million training steps, the learning curves in Fig. 5 do not have particularly steep lines, but are rather flat. Is the training environment valid for tracking tasks?

**Clarity:**

The paper is straightforward and well-written. However, I regret that certain points, such as the communication mechanism (TarMAC, I2C) in section 5, were not understood. Also difficult to comprehend completely is section 5.3. It would be courteous to display additional relevant work for the paper in section 2.

**Correctness:**

The structure of the state, observation, action, entities, and reward looks well constructed. But seeing Fig 5(b), the communication property seems to not affect training. It makes me wonder whether the individual and public observation or states are well constructed or not.

**Documentation:**

Well documented.

**Ethics:**

Well explained about the concerns on social impact.

**Relation To Prior Work:**

The authors mentioned the prior work focusing on the Multi-Camera Multi-Target Tracking.

**Summary And Contributions:**

This paper introduces a new dataset for multi-agent reinforcement learning in tracking tasks. The authors present not only tracking tasks but also tasks conveyed by mobile agents, enabling cooperative, competitive, and mixed games to be played with ease. Contributions are as follows:
 * Multi-Agent tracking tasks combined with the shipments tasks carried by mobile agents with properties like communications, heterogeneous, and asymmetric composition of agents.
 * Cooperative, Competitive and mixed tasks
 * Easy to use, requiring low memory for CPU, which enables sample efficiency by distributed experiments.
 * Can present researchers with a basic environment for multi-camera tracking tasks.

---

> ### Author Response · Authors · 2022-08-22
> **Response to Reviewer 4Kzr (4/4)**
>
> > **AF8:** Do you have any experience with curriculum learning for training the algorithms in Figure 5 that were not trained?
>
> **A8:**
> We believe it is strongly viable to construct a curriculum relative to varying the number of agents in the environment—for example, 4C vs. 2T, 4C vs.4T, and 4C vs. 8T. The steeper convergence in the training curves (Fig.G.1 in the supplementary, or [Exhibit C](https://sites.google.com/view/mate-neurips2022/home)) appears in the easier scenarios (e.g. 4C vs. 2T).
>
> ------
>
> > **AF9:** In section 5, the HRL model receives a high-level signal from the MARL algorithm and selects a rule-based action. However, since the signal may be a latent vector, isn't it necessary to train the HRL model with the signal as an input?
>
> **A9:**
> In our HRL model, only the high-level policy is trained, while the low-level policy is rule-based. The high-level policy takes local observation as well as messages received through communication. The policy will output a boolean array, i.e., the selection, telling the low-level policy which targets to track. The low-level policy does not require a latent vector signal but an interpretable boolean array with the length of the number of targets.
>
> ------
>
> > **AF10:** I want to know the selection criteria for baseline algorithms (IPPO, MAPPO, QMIX, MADDPG). There are SOTA multi-agent algorithms that are policy gradient and value-based in cooperative and competitive environments.
>
> **A10:**
> We pick MARL algorithms based on the following criteria: value-based/policy-based, on-policy/off-policy, centralized-training/decentralized-training properties. We added a new table in the manuscript (or view [Exhibit D](https://sites.google.com/view/mate-neurips2022/home)) appears in the easier scenarios (e.g. 4C vs. 2T).) to demonstrate the properties of the baseline algorithms in the experiments.
>
> ------
>
> > **W2:** Baselines are not enough to show the validity of the environment, as shown in Fig 5. Lines are so flat that it doesn't looks like being under training for 10 million steps.
> > **AF11:** I have a strong suspicion regarding the training camera experiments in the fully cooperative game. Despite 10 million training steps, the learning curves in Fig. 5 do not have particularly steep lines, but are rather flat. Is the training environment valid for tracking tasks?
>
> **A2 & A11:**
> We are afraid that you might be referring to the two subfigures of Fig. 4 instead of Fig. 5? Because Fig. 5 shows the training curves for the target agents, and in our opinion, these curves are quite steep. Regarding Fig.4, it is one of the more challenging scenarios because it is 4 camera agents competing against 8 target agents with 9 obstacles present in the environment. In the 4C vs. 2T scenario (Fig. 5), the baseline algorithms can achieve a high target coverage rate with only a small number of samples. As you can see from Fig.G.1 in the supplementary (or [Exhibit C](https://sites.google.com/view/mate-neurips2022/home)), the difficulty level for training the cameras progresses with the number of target agents. We test the baseline algorithms for the camera agents in the 4C vs. 8T scenario. It is comparatively a harder setting for the camera agents because the number of targets is twice as many as the cameras, so it is difficult to cover all the targets simultaneously. We test the baseline algorithms in the 4C vs. 2T,  4C vs. 4T, and 4C vs. 8T scenarios. The result shows steeper convergence on the training curves in the easier scenarios (Fig. G.1 in the supplementary, or [Exhibit C](https://sites.google.com/view/mate-neurips2022/home)).
>
> Difficult scenarios will lead to more high-variance trajectories filled with low-reward experiences, making the credit assignment harder. Additionally, camera agents have partial observability over the environment, and their spawn locations are randomized within a certain range at the beginning of every episode. More importantly, in the fully-cooperative game, all camera agents will share the same reward. This is prone to difficult credit assignments due to the “lazy” agent problem and non-stationarity environment dynamics, which are similarly described in the VDN literature [3]
>
> ------
>
> References:
>
> [1] Lanctot, Marc, et al. "A unified game-theoretic approach to multiagent reinforcement learning." Advances in neural information processing systems 30 (2017).
>
> [2] Irpan, Alex. ‘Deep Reinforcement Learning Doesn’t Work Yet’. N.p., 2018. Web.
>
> [3] Sunehag, Peter, et al. "Value-decomposition networks for cooperative multi-agent learning." arXiv preprint arXiv:1706.05296 (2017).

---

> ### Author Response · Authors · 2022-08-22
> **Response to Reviewer 4Kzr (3/4)**
>
> > **AF5:** What effect does 'minimizing the exposure time to cameras' have on target agents in terms of real-world applications?
>
> **A5:**
> Our original intent for target-tracker coevolution is geared toward building a generalizable multi-agent tracking system via adversarial training. As described in the introduction, multi-agent tracking systems have many potential use cases in real-life scenarios. We depict the target agent as the “evil” counterpart that relentlessly searches for weaknesses in the collaborative target coverage strategy.
>
> ------
>
> > **AF6:** Env SMAC is written to be non-scalable in table 1, but I believe it is scalable because it can produce more units depending on the settings.
>
> **A6:**
> Sorry for the inconvenience. Our criteria for scalability is that the user can change the agent quantity in the environment without additional effort. In SMAC, the user needs to **make a new map with the StarCraft II Map Editor to have a different number of agents**. That is non-trivial extra work. In MATE, the number of agents can be configured in the configuration files or through the Python APIs. We also provide a procedural generator to create a game scenario with the given configuration by the user (a demo can be found in our supplementary video file, 0:38-0:46 ,or at the bottom of our [Google Site](https://sites.google.com/view/mate-neurips2022/home)). While there is not a procedural generator for maps in SMAC to our knowledge. Besides, the MATE environment supports up to hundreds of agents interacting with the environment simultaneously.
>
> ------
>
> > **AF7:** In Figure 3, I wonder how to choose between peer-to-peer and broadcast methods for passing messages. How do agents choose a peer when using the peer-to-peer method?
>
> **A7:**
> There are many existing works for multi-agent communication. The choice of communication modes is depended on the algorithm. For example, The TarMAC uses the broadcast method for communication and the I2C (short for “Individually Inferred Communication”) algorithm chose peer-to-peer communication in the experiment. In I2C, each agent learns a prior network to choose which partner to communicate with.
> The communication edges are inferred by “causal inference” in the paper. An agent is more likely to communicate with the partner whoever has more effect on its action. Formally, for current agent $i$ and its partner $j$:
>
> $$
> \operatorname{Prior} (i, j) = \mathbb{I} [ D_{KL} ( P (a_i | a_{-i}, o_i) \Vert P (a_i | a_{-ij}, o_i) ) \ge \delta \ ]
> $$
>
> where the distributions are learned by soft-learning:
>
> $$
> P (a_i | a_{-i}, o_i) = \frac{\exp ( \lambda Q (a_i, a_{-i}, o_i) )}{\sum_{a_i'} \exp ( \lambda Q (a_i', a_{-i}, o_i) )}
> $$
>
> $$
> P (a_i | a_{-ij}, o_i) = \sum_{a_j} P (a_i, a_j | a_{-ij}, o_i)
> $$
>
> and the $Q$ is the joint state-action value function, and $\lambda$ is a hyperparameter.
>
> The agent $i$ chooses to communicate with agent $j$ if:
>
> $$
> \operatorname{Prior} (i, j) = \mathbb{I} [ D_{KL} ( P (a_i | a_{-i}, o_i) \Vert P (a_i | a_{-ij}, o_i) ) \ge \delta \ ]
> $$
>
> In the I2C paper, the communication message is the local observation of agent $j$.
> Note that the problem of whom to communicate with is underexplored in the studies of multi-agent communication, yet peer-to-peer communication exists in many aspects of real-life applications. This motivated us to implement the peer-to-peer communication mechanism in MATE to provide a benchmark for this field.

---

> ### Author Response · Authors · 2022-08-22
> **Response to Reviewer 4Kzr (2/4)**
>
> > **W3:** So short Related Work section that I felt hard to understand in some sections like 5.3.
>
> **A3:**
> Sorry for the inconvenience. We updated the related works and expanded on Section 5.3 accordingly. We added more references regarding game theory, self-play, auto-curriculum, and multi-agent communication to help readers understand the experiment section of our manuscript.
>
> Here we provide a summary of Section 5.3 for your convenience. Previous works suggest that training agents against fixed-strategy opponents will likely lead to failure in generalization (as in playing against various opponents) [1][2]. Instead of designing a limited set of rule-based motion patterns for targets with human knowledge, we propose to train both trackers and targets in coevolution with a zero-sum payoff structure, facilitating the process of autocurricula. We decided on the zero-sum game setting because it is strictly competitive, and trackers and targets must both learn how to exploit each other to the full extent.
>
> The best performing baseline, the Policy-Space Response Oracle (PSRO) [1] with Nash Equilibrium solver, computes a meta-game strategy profile at the end instead of a single policy in the fully-cooperative game setting in Sections 5.1 and 5.2. The X-axis of Fig. 7 (and maybe 8) are population iteration, meaning that the population-based training must undergo multiple training rounds as opposed to a single round in Sections 5.1 and 5.2. At each population iteration, we train a deep-RL policy to convergence that approximates the best response (BR) against the opponent population. After we obtain this BR, it is appended to the policy memory, and we use a Nash Equilibrium solver to find a mixed-strategy nash equilibrium given this set of policies as all possible strategies. This entire process of BR computation and solving for nash equilibrium is sequentially iterated between cameras and targets. You may find the pseudocode of Algorithm 1 in [the PSRO paper](https://proceedings.neurips.cc/paper/2017/file/3323fe11e9595c09af38fe67567a9394-Paper.pdf) helpful for understanding. Solving BR iteratively using a deep RL algorithm is backed by the convergence analysis of the Double Oracle algorithm mentioned in the same literature.
>
> On the other hand, fictitious self-play (FSP) uniformly samples responses from the policy memory. Therefore if we were to draw the policy distribution for FSP in Fig. 8, it would look like a uniform distribution. In comparison, the policy distribution computed by PSRO-Nash looks more dispersed.
>
> Please feel free to let us know if there is anything else that requires further elaboration.
>
> ------
>
> > **C4:** The structure of the state, observation, action, entities, and reward looks well constructed. But seeing Fig 5(b), the communication property seems to not affect training. It makes me wonder whether the individual and public observation or states are well constructed or not.
>
> **A4:**
> For the camera agents trained using the HRL algorithm, we observe that the addition of a communication module has a negligible impact on the performance. We think that is due to the inductive bias introduced in the hierarchical policy. In the HRL policy, there is a higher-level “manager” sub-policy to select a single or multiple tracking targets for the lower-level “executor” sub-policy based on mean total coverage, and the executor policy is a rule-based program that rigorously tracks the target whichever asked by the manager policy. In the setting of our experiment, communication only happens among the higher-level policies, which is reasonable because we argue that a lower-level policy should not be subjected to external influences other than its overlord. While under partial observability, a higher-level policy may receive the locations of out-of-sight targets from its peers. However, the underlying low-level executor is prohibited select to track these targets that are not present in its local observation. This explains why the HRL method did not benefit from utilizing communication.
> For the other non-hierarchical methods in our experiments, there is only a single layer of execution, in which the camera agents output the view-controlling actions (rotating and zooming) directly. Training with the communication add-on will facilitate better decision-making, therefore improving performance. Also, we have demonstrated the impact of partial/full observation on algorithm learning in the experiments of the target agents (Fig. 6, Full vs. Partial). The observability mode has an increasingly significant impact when there are more agents in the environment.

---

> ### Author Response · Authors · 2022-08-22
> **Response to Reviewer 4Kzr (1/4)**
>
> Thank for your insightful comments. Please kindly refer to the [Common Response](https://openreview.net/forum?id=SyoUVEyzJbE&noteId=GbmiOtZi_G0) at above for the list of changes in our manuscript and supplementary material, and a dedicated website ([link](https://sites.google.com/view/mate-neurips2022)) for our rebuttal. We look forward to further discussion.
>
> > **W1:** The GUI closely resembles Multi-Agent Particle Environment (MPE), which contradicts the authors' claims of realism. Therefore, it is insufficient for authors to advocate realistic characteristics.
>
> **A1:**
> MATE is designed to resemble the target coverage problem in real-life applications. Many realistic applications we found in real-life (e.g., anti-poaching, border security, ball-tracking) relies on reliable strategies for coordinating multiple cameras to achieve users' goals.
>
> Also, the core aspect of the target coverage problem is coordinating multiple directional sensors (camera) for maximum target coverage. This is a non-trivial and yet-to-be-solved problem in the MARL domain. As you can see from Fig. 4, all of the multi-agent algorithms benchmarked did not outperform our rule-based control system in target-dominating scenarios. In order to accelerate the training of multi-agent policies, we applied a minimalistic aesthetic in MATE to simplify the computation of the perception model. Adding more visual elements (e.g., 3D perception) to the environment will make our application look more photorealistic, but it does not change the fact that the basis of the problem still concerns the coordination strategy of multiple cameras against versatile targets.
> It is why we claimed that the MATE is realistic in terms of the setting of the research problem, while other factors that contribute to photorealism are less prioritized in our research before reliable coordination is established.
>
> Compared with the Multi-Agent Particle Environment (MPE), the entities in MATE have partial observability rather than that in the original MPE, which has full observability over the entire environment. For example in MATE, the cameras cannot observe target agents behind an obstacle (if transmittance is set to 0%). Besides, the obstacle in MATE is a hard-restricted area that the target agents cannot run into. In MPE, obstacles only apply an extra soft force to the agent dynamic, sometimes resulting in agents clipping into the obstacle (See 0:32, https://www.youtube.com/watch?v=sSltKKwCXbM&ab_channel=VictorGouet ).

---

> ### Author Response · Authors · 2022-08-27
> **Looking forward to discussions**
>
> Dear Reviewer 4Kzr,
>
> We deeply appreciate your time and effort to evaluate our submitted work. We were just wondering if our rebuttal to your comments has successfully addressed your concerns. In case you have any remaining concerns please do not hesitate to discuss them with us! Thanks in advance.
>
> --- Authors of Paper66

---

### Official Review · Reviewer_jDDc · 2022-07-28
**Review of "MATE: Benchmarking Multi-Agent Reinforcement Learning in Distributed Target Coverage Control"**

**Rating:** 6
**Confidence:** 4

**Strengths:**

This paper introduces an interesting and novel learning environment. It incorporates a few features that I have not seen combined in other environments, including teaming, partial observability, and non-zero-sum reward. The code is compatible with the standard OpenAI Gym interface and with RLLib. It has the potential to open up new research directions in algorithm development for this setting.

**Weaknesses:**

- The entire setup is an important step towards a general learning environment for studying this problem, but presently quite narrow in scope. For example, lines 139-141 state that "there are two kinds of vehicles for targets, one with high speed and small capacity, and the other with low speed and large capacity. The former is as twice fast as the latter but with a halved carrying capacity" - why not have parameters for speed, capacity, size, etc.? Why not let the agents set their own speed via an action? These suggestions are about this as a learning environment that others can use, as opposed to a single stylized simulation for a specific learning task.
- Table 1, while containing helpful comparisons, can be misleading in the conclusions that it implies by its use of colors (green implying good). Specifically, it appears to imply that the MATE environment is the only environment that has all of the desirable features compared to the other environments listed. The more accurate conclusion that should be drawn from this table is that, along the indicated axes, MATE has a unique set of attributes relative to other learning environments. However, given its relatively narrow focus, it should be clearer that MATE does not supersede these environments but rather provides a new learning environment with a novel set of features.
- The "obstacle" defined in Section 3.1 is really an environment feature. A more general version of this environment would allow arbitrary obstacles and layouts with different topologies.
- Depending on whether the authors would expand the Experiments section, this may be a more appropriate submission for the NeurIPS Main Track than the Datasets & Benchmarks track.

**Additional Feedback:**

Minor comments:
- Line 197: it would be helpful to be concrete about what "modern CPU" means (especially if this paper is read in the future)

**Clarity:**

The paper is written clearly. There are some sections that could be rephrased but everything is generally easy-to-read.

**Correctness:**

There are no concerns about correctness, although I am not specifically evaluating the experiments from Section 5 (treating them more as a proof-of-concept that this learning environment can be used for training agents in different scenarios). I would suggest deemphasizing this section and putting more emphasis on how users of this learning environment can expand it and provide modifications to the base scenario.

**Documentation:**

The GitHub repository is clear with good documentation. There could be more details provided on setup and getting started in different OSes.

**Ethics:**

While the topic of the paper, a reinforcement learning environment for tracking of mobile targets, can potentially raise some concerns, this paper itself just provides a stylized environment that is unlikely to be used in practical applications. The authors mention this in Section 6 of the paper.

**Relation To Prior Work:**

The related work section (Section 2) is quite narrow in scope. Especially in the context of learning environments, there could be more detail on what this kind of learning environment provides.

**Summary And Contributions:**

This paper introduces MATE, a multi-agent learning environment that models an asymmetric game between two teams of players: the "cameras" and the "targets". The cameras are fixed in their location but can change direction and zoom, with the goal of observing the targets. Their sensors can not see through obstacles in the environment. The targets are mobile agents with sensors that can see through obstacles, with the goal of moving items to warehouses. In terms of code, this environment provides an OpenAI Gym interface for interacting with the environment and training agents.

---

> ### Author Response · Authors · 2022-08-22
> **Response to Reviewer jDDc (2/2)**
>
> > **W2.1:** Table 1, while containing helpful comparisons, can be misleading in the conclusions that it implies by its use of colors (green implying good). Specifically, it appears to imply that the MATE environment is the only environment that has all of the desirable features compared to the other environments listed. The more accurate conclusion that should be drawn from this table is that, along the indicated axes,
>
> **A2.1:**
> Thanks for the suggestion. We updated the manuscript accordingly. We recognize that the previous use of colors in Table 1 could mislead the readers, therefore we have changed them to more neutral colors.
>
> > **W2.2:** MATE has a unique set of attributes relative to other learning environments. However, given its relatively narrow focus, it should be clearer that MATE does not supersede these environments but rather provides a new learning environment with a novel set of features.
>
> **A2.2:**
> To clarify, we do not believe that MATE would either supersede or replace other multi-agent environments in their entire usage. But we genuinely believe that MATE can function as a good benchmark for the target coverage control problem as well as the more general multi-agent learning benchmark.
>
> Regarding the contribution of MATE as a new multi-agent environment, we present a new benchmark that concerns several prominent research topics in the study of multi-agent learning, such as multi-agent communication/negotiation, asymmetric game, multi-agent credit assignment, scalability, opponent modeling, and decentralized coordination. These research points raised in MATE will similarly exist in many real-world applications. Thus, we argue that benchmarking the off-the-shelf MAL algorithms in MATE can expose and project possible weaknesses in deployment and push the community toward studying more practical algorithms for real-world applications.
>
> Regarding the contribution of MATE as a novel environment for studying the target coverage control problem, it provides an easy-to-use simulation for training the learning-based methods and evaluating the multi-agent tracking system on coverage rate and communication efficiency. The procedural generator helps the users generate new environments with different configurations. MATE has environmental features (e.g., procedural generation of scenarios, random obstacles) and task features (e.g., transport task, bounty reward) to encourage dynamic competition between the two rival groups and to allow the emergence of more diverse and interesting strategies. The controllable targets can enable the adversarial training framework to build more robust control policies for the tracking systems.
>
> ------
>
> > **W3:** The "obstacle" defined in Section 3.1 is really an environment feature. A more general version of this environment would allow arbitrary obstacles and layouts with different topologies.
>
> **A3:** We represent all objects in a circular shape to simplify the computation and the observation representation. The location and radius of all obstacles are fully configurable by the user. The circle objects can be regarded as the atoms of the world in MATE, where we can fit different topologies with many small dots. For example, the user can put multiple circles in a line to use as a “wall”. We have prepared more examples of obstacle shapes in [Exhibit B](https://sites.google.com/view/mate-neurips2022/home)
> Later we will provide scripts to help users implement arbitrary obstacles in more complex shaping in a future version.
>
> ------
>
> > **W4:** Depending on whether the authors would expand the Experiments section, this may be a more appropriate submission for the NeurIPS Main Track than the Datasets & Benchmarks track.
>
> **A4:**
> Thanks for your consideration. The main contribution of this paper is designing an “all-in-one” multi-agent tracking environment for benchmarking the multi-agent learning algorithms. We conduct experiments on MATE to show the feasibility of our design and to showcase the existing gap between multi-agent learning and real-world applications in terms of the target coverage problem. However, we did not intend to contribute any new algorithms in this paper. The MARL algorithms and population-based training methods mentioned in this paper are from previous works and are adequately referenced. Therefore we argue that our paper would be more appropriate for the Datasets & Benchmarks track.
>
> ------
>
> > **AF5:** Line 197: it would be helpful to be concrete about what "modern CPU" means (especially if this paper is read in the future)
>
> **A5:**
> Sorry for the ambiguity; we updated the manuscript as you suggested. We tested our environment using a single-threaded program with an Intel i7 8700 @ 3.20GHz CPU.

---

> ### Author Response · Authors · 2022-08-22
> **Response to Reviewer jDDc (1/2)**
>
> Thank for your insightful comments. Please kindly refer to the [Common Response](https://openreview.net/forum?id=SyoUVEyzJbE&noteId=GbmiOtZi_G0) at above for the list of changes in our manuscript and supplementary material, and a dedicated website ([link](https://sites.google.com/view/mate-neurips2022)) for our rebuttal. We look forward to further discussion.
>
> ------
>
> > **W1:** The entire setup is an important step towards a general learning environment for studying this problem, but presently quite narrow in scope. For example, lines 139-141 state that "there are two kinds of vehicles for targets, one with high speed and small capacity, and the other with low speed and large capacity. The former is as twice fast as the latter but with a halved carrying capacity" - why not have parameters for speed, capacity, size, etc.? Why not let the agents set their own speed via an action? These suggestions are about this as a learning environment that others can use, as opposed to a single stylized simulation for a specific learning task.
>
> **A1:**
> Thanks for the suggestions. The vehicle capacity, size, and max speed are already parameterized, and these parameters are customizable in the provided environment configuration file. The user can also customize the size of the camera’s detection zone and rotational and zooming speed. Regarding the specific speed of target agents, they have continuous or multi-level discretized action space as shown in [Exhibit A](https://sites.google.com/view/mate-neurips2022/home) appears in the easier scenarios (e.g. 4C vs. 2T).(Details refer to here: [Doc Page 1](https://mate-gym.readthedocs.io/en/latest/environment/actions.html), [Doc Page 2](https://mate-gym.readthedocs.io/en/latest/wrappers.html)). Thus, the target agents can set their speed by changing the magnitude of their actions. The speed setting in the configuration file controls the maximum speed of vehicles. The user can configure different max speeds for every type of vehicle.

---

> ### Author Response · Authors · 2022-08-27
> **Looking forward to dicussions**
>
> Dear Reviewer jDDc,
>
> We deeply appreciate your time and effort to evaluate our submitted work. We were just wondering if our rebuttal to your comments has successfully addressed your concerns. In case you have any remaining concerns please do not hesitate to discuss them with us! Thanks in advance.
>
> --- Authors of Paper66

---

### Author Response · Authors · 2022-08-22
**Common Response**

## Revision Notes

We have updated the manuscript and the supplementary material. For the convenience of the reviewers and area chair, we have created a dedicated Google Site containing graphics, tables, and videos mentioned in our rebuttal ([link](https://sites.google.com/view/mate-neurips2022)).

Here is the list of changes:

**Manuscript:**

1. Added more content and related works regarding the target coverage control problem.
1. Added more content and related works regarding game theory, self-play, and population-based training regimes.
1. Added footnotes to clarify “modern CPU” and “minimal dependencies”
1. Added a table to list the properties of baseline algorithms in the experiment section
1. Further expanded on Section 5.3 to help readers understand our approach to training agents in zero-sum games.

**Supplementary Material:**

1. Performed experiments regarding training camera agents at different difficulty levels. Add the training curves for comparison.
1. Added new baseline results for the HiT-MAC algorithm to train camera agents.

Text changes in the manuscript are colored in blue.

---

### Meta-Review · Area_Chair_2xX1 · 2022-09-12

**Recommendation:** Accept
**Confidence:** 4

**Metareview:**

This paper introduces an interesting cooperative-competitive MARL environment involving groups of camera and targets with the objective of transportation. The benchmark environment appears to be scalable with novel components including partial observability and general-sum rewards in an interesting environment. The authors provide extensive baselines for MARL algorithms as well. The majority of reviewers appraised the work positively, and the authors adequately addressed the concerns brought up in the review along with a revision incorporating some components of feedback. I think this is a good addition to the MARL literature, and the authors should be sure to incorporate all the remaining feedback in the final version of the paper.

---

### Decision · Program_Chairs · 2022-09-16

Accept